# Handheld Inkjet Printing Paper Chip Based Smart Tetracycline Detector

**DOI:** 10.3390/mi10010027

**Published:** 2019-01-01

**Authors:** Jiahao Li, Xin Wang, Yanke Shan, Huachuan Huang, Dan Jian, Liang Xue, Shouyu Wang, Fei Liu

**Affiliations:** 1Joint International Research Laboratory of Animal Health and Food Safety of Ministry of Education & Single Molecule Nanometry Laboratory (Sinmolab), Nanjing Agricultural University, Nanjing 210095, Jiangsu, China; lijiahao690940@sina.com (J.L.); 18168021002@163.com (X.W.); shanyk26@126.com (Y.S.); 2School of Manufacturing Science and Engineering, Southwest University of Science and Technology, Mianyang 621010, Sichuan, China; hhc425@163.com; 3Computational Optics Laboratory, School of Science, Jiangnan University, Wuxi 214122, Jiangsu, China; jiandancx@sina.com; 4College of Electronics and Information Engineering, Shanghai University of Electric Power, Shanghai 200090, China; xueliangokay@gmail.com

**Keywords:** smart tetracycline detector, inkjet printing paper chip, on-site detection

## Abstract

Tetracycline is widely used as medicine for disease treatments and additives in animal feeding. Unfortunately, the abuse of tetracycline inevitably causes tetracycline residue in animal-origin foods. Though classical methods can detect tetracycline in high sensitivity and precision, they often rely on huge and expensive setups as well as complicated and time-consuming operations, limiting their applications in rapid and on-site detection. Here, we propose a handheld inkjet printing paper chip based smart tetracycline detector: tetracycline can be determined by inkjet printing prepared paper chip based enzyme-linked immunosorbent assay (ELISA) with the advantages of high sensitivity, excellent specificity and low cost; moreover, a smartphone based paper chip reader and application is designed for automatically determining tetracycline with simple operations, high precision and fast speed. The smart tetracycline detector with a compact size of 154 mm × 80 mm × 50 mm and self-supplied internal power can reach a rather low detection limit of ~0.05 ng/mL, as proved by practical measurements. It is believed the proposed handheld inkjet printing paper chip based smart tetracycline detector is a potential tool in antibiotic sensing for routine uses at home and on-site detection in the field.

## 1. Introduction

Antibiotics are not only commonly used as medicine for disease treatments, but also are widely used in additives for animal feeding. Among various antibiotics, tetracycline is the one most widely used as a feed additive for animal disease prevention and treatment due to its wide antimicrobial capability, low cost and good stability. Unfortunately, the abuse of tetracycline, especially in animal feeding, inevitably causes tetracycline residue in animal-origin foods such as meat, eggs and milk, etc., not only causing adverse reactions including abdominal pain, diarrhea, flatulence and vomiting, but also damaging human organs such as the liver and kidney, and even increasing the risk of bacterial resistance. As such, it is important to monitor tetracycline, especially in animal-origin foods [1,2,3].

Various methods have been proposed to detect tetracycline, such as high performance liquid chromatography (HPLC) [4,5,6], liquid chromatography-mass spectrometry (LC-MS) [7,8,9], microbiological testing [10,11], capillary electrophoresis [12,13], colloid gold immunochromatography assay (CGIA) [14,15,16] and enzyme-linked immunosorbent assay (ELISA) [17], etc. HPLC and LC-MS can detect tetracycline in rather high precision and sensitivity, but they require complicated sample pretreatment and time-consuming operations, limiting their applications in rapid detection. While microbiological testing can detect tetracycline with simpler operations and faster speed compared to HPLC and LC-MS, its detecting specificity is poor. Moreover, these methods often rely on huge and expensive instruments, not only increasing the detecting cost, but also limiting their on-site detecting applications. Tetracycline detection based on CGIA has the advantages of fast speed and simple operation, which make it a potential approach for on-site detection, unfortunately it still suffers from low sensitivity and high cost. ELISA is widely used in tetracycline detection due to its high specificity and sensitivity; however, it is often time-consuming and expensive, and it still relies on huge and expensive equipment such as a microplate reader, therefore limiting its potential applications in rapid on-site screening for a large number of samples [18]. In order to reduce costs, paper chip based ELISA testing was proposed since it not only maintains the advantages of traditional ELISA such as high specificity and sensitivity, but also obviously reduces the detection cost [19,20,21,22]. Moreover, paper chip based ELISA even avoids the expensive equipment reliance, since the detecting results can be simply determined using the naked eye. In recent years, a growing number of works have reported the immobilization of the antibody on the paper chip using the chitosan-glutaraldehyde crosslinking method, which first requires chitosan to interact with the paper chip for 30 min, and then takes ~2 h to crosslink glutaraldehyde with chitosan. Therefore, these chitosan-glutaraldehyde crosslinking fabricated paper chips often require extremely long time-consuming preparation [23,24].

In order to further accelerate the detecting speed of paper chip based ELISA as well as decrease its detecting cost, we design a handheld inkjet printing paper chip based smart tetracycline detector. First, we propose low-cost paper chips performed by simple and cost-effective inkjet printing method [25,26,27]. Then, tetracycline detection is realized by paper chip based ELISA. The tetracycline complete antigen is fixed on the paper chip after being processed by the plasma ultrasonic cleaner, then tetracycline complete antigen competes with the tetracycline from the sample by binding to the anti-tetracycline monoclonal antibody. Finally, tetracycline is determined by tracing the signal tag on HRP-Ab2: the peroxidase enzyme can oxidize 3,3′,5,5′-tetramethylbenzidine (TMB) by H_2_O_2_, causing the color change. In order to precisely determine the color change, we design a smartphone based paper chip reader and the corresponding application for automatically detecting tetracycline, and similar to other smartphone based miniature mobile devices [28,29,30,31,32,33], the smartphone based paper chip reader and application can provide rapid and precise measurements. Combined with the inkjet printing paper chip for ELISA testing and smartphone based paper chip reader for data collection and analysis, the handheld inkjet printing paper chip based smart tetracycline detector has a rather compact size of 154 mm × 80 mm × 50 mm and self-supplied internal power with the rather low detection limit of ~0.05 ng/mL, as proved by practical measurements. Moreover, the operations are simple and can be finished within 40 min, and the cost of the inkjet printing paper chip for single detection is below 1 RMB. Compared to a commercial ELISA kit, the proposed handheld inkjet printing paper chip based smart tetracycline detector has similar sensitivity, but has a much faster speed and lower cost. Considering its advantages including compact configuration, high sensitivity, fast speed, simple operation and low cost, it is believed the proposed handheld inkjet printing paper chip based smart tetracycline detector is a potential tool in antibiotic sensing for routine uses at home and on-site detection in the field.

## 2. Materials and Methods

### 2.1. Chemicals and Reagents

Trimethoxyoctadecylsilane (C_21_H_46_O_3_Si), bovine serum albumin (BSA) and tetracycline hydrochloride (C_22_H_24_N_2_O_8_·HCl) were purchased from Sigma-Aldrich (St. Louis, MO, USA). Ampicillin, penicillin, streptomycin, kanamycin, dipotassium phosphate (K_2_HPO_4_), monopotassium phosphate (KH_2_PO_4_), sodium chloride (NaCl) and magnesium chloride (MgCl_2_) were purchased from Sinopharm (Hong Kong, China). Tween-20 was purchased from Solarbio (Beijing, China). The filter paper was purchased from Whatman (Maidstone, UK). Tetracycline-BSA antigen and anti-tetracycline monoclonal antibody were purchased from Suzhou Kuaijiekang Co., Ltd. (Suzhou, China). Commercial tetracycline ELISA kit was purchased from Shenzhen Finder Biotech Co., Ltd. (Shenzhen, China). The heating plate (YH-946B) was purchased from Shanghai Yuxing Electronics Co., Ltd. (Shanghai, China). The inkjet printer (IP2780) was purchased from Canon (Tokyo, Japan).

### 2.2. Milk Samples

3 brands of milk purchased from Nanjing local supermarket were tested by LC-MS (QTRAP 6500, Framingham, MA, USA), indicating that no tetracycline had developed in the milk samples. Different concentrations of tetracycline were added in these tetracycline-free milk to simulate the practical samples.

### 2.3. Inkjet Printing Paper Chip Preparation

Paper chips were prepared according to the inkjet printing method. First, the paper chip with both the test and control regions with the diameter of 6 mm and the separation of 6 mm was designed. If the size of the paper chip is further decreased, while the volume of the solutions for target detection can be reduced, the interval between the control and test regions will be even smaller, as cross-interaction inevitably occurs during the sampling adding and washing procedures. Meanwhile, the large size of the paper chip not only requires more sample solution, but also leads to inhomogeneous coloration. We optimized the size of the paper chip, and found the used configuration not only had homogeneous coloration, but also avoided the cross-interaction during sample adding and washing. In addition, it required relatively a low volume of the solution. Then the hydrophobicity reagents (2% N-heptane solution of octadecyl trimethoxysilane) were printed on the filter paper using the inkjet printer. Next, the filter paper was heated using the heating plate at 100 °C for 90 min, and set at room temperature for 3 h. Finally, the filter paper was treated using oxygen plasma cleaner for 4 min (the frequency of plasma was 13.56 MHz and the power was 100 W) to prepare the inkjet printing paper chips in order to generate functional aldehyde group on the paper chip surface for protein immobilization. The cost of the inkjet printing paper chip for single detection is below 1 RMB.

### 2.4. Paper Chip based ELISA Testing

The inkjet printing paper chip based ELISA testing process is illustrated in Figure 1. First, 50 μL tetracycline complete antigen was added to both the test and control regions modified by aldehyde group and incubated for 15 min at room temperature. Unreacted antigen was washed off with 40 μL of 1% PBST buffer. Here, a conventional washing method was implemented, in which a piece of filter paper was placed under the paper chip to absorb excess buffer solution [21,34]. Then, 50 μL of 1% BSA solution was added to both the test and control regions for 15 min to block unreactive binding sites on the paper chip and reduce nonspecific adsorption. Afterwards, the same washing method was implemented to wash off the unreacted BSA. Next, the sample under detection and the control buffer without tetracycline were added to the test and control regions, respectively, and 100 μL anti-tetracycline monoclonal antibody solution with the concentration of 8 μg/mL was simultaneously added to both regions. After 5 min incubation, the unbinding anti-tetracycline monoclonal antibody and tetracycline were washed off with 40 μL of 1% PBST buffer. Finally, 10 μL of goat-anti-mouse IgG labeled with HRP was added to both the test and control regions and incubated for 5 min to form antigen-antibody-enzyme-labeled antibody complex. After 40 μL of 1% PBST buffer was used to wash off unreacted enzyme-labeled antibodies, the paper chip was directly imaged for tetracycline detection. The inkjet printing paper chip based ELISA testing could be finished within 40 min, indicating a rather fast processing speed.

### 2.5. Tetracycline Detection using Commercial ELISA Kit

Tetracycline in PBS and milk were detected using the commercial ELISA kit (Shenzhen Finder Biotech Co., Ltd. China). Note that the milk sample should be pre-treated according to the instructions of the commercial ELISA kit before detection. The sample was tested according to the following steps. (1) Add 50 µL of target solution to the microwells and another 50 µL of the primary antibody solution (anti-tetracycline monoclonal antibody), and then incubate at 25 °C for 30 min. (2) Pour the liquid out from the microwells and add 250 µL/well of washing buffer for 15–30 s, repeat them for 5 times, and then flap to dry. (3) Add 100 µL solution of the secondary antibody (goat-anti-mouse IgG) labeled with HRP and incubate at 25 °C for 30 min, and then wash the microwells with the same procedures as (2). (4) Add 50 µL of tetramethylbenzidine (TMB) substrate A and 50 µL of the TMB substrate B into each microwell followed by incubation at 25 °C for 15 min. (5) Add 50 µL of the stop solution into each microwell, and set the wavelength of the microplate reader (Multiskan FC, Thermo Fisher, Waltham, MA, USA) at 450 nm to measure the OD values for tetracycline detection.

## 3. Results

### 3.1. Smartphone Based Paper Chip Reader and Application

Figure 2A shows the smartphone based paper chip reader. A white-light LED (Shenzhen Jiafurui Lighting Co., Ltd., Shenzhen, China) supplied by two AA batteries was adopted to illuminate the paper chip, which was directly imaged by smartphone camera (Nubia, Shenzhen, China). In order to compress the size of the imaging system, a flat convex lens (Daheng Optics, Shanghai, China) with the focal length of 30 mm was used as a relay lens. A 3-D printed shell was fabricated to integrate the white-light LED array, the relay lens and the smartphone as shown in Figure 2B, and the paper chip was fixed by a holder connected with the imaging system through magnet. The assembled hand-held smartphone based paper chip reader can be adopted for tetracycline detection.

In order to determine tetracycline in high sensitivity and precision, as well as with fast speed and simple operation, a smartphone application called tetracycline colorimetry was designed as shown in Figure 3. First, after clicking the icon on the smartphone desktop in Figure 3A, the application was started up and its initial interface is shown in Figure 3B. Then, to use the application for tetracycline detection, click the “START” button for the paper chip image input in Figure 3C. Afterwards, click the “TAKE PHOTO” button to call the smartphone camera application for image recording or click the “CHOOSE FROM ALBUM” button to read-in the captured image. Next, the quality of the captured image was automatically checked by clicking the “CHECK” button in order to guarantee the exposure time was correctly selected to pursue high signal to noise ratio but to avoid over-exposure as shown in Figure 3D. Both the test and control regions were recognized through threshold segmentation. In order to obtain a high signal to noise ratio, the average intensity in the control region in the blue channel should be no less than 150, and the pixel intensity should not reach 255 for the captured 8-bit image. These two criteria should be both satisfied, otherwise another image should be inputted for precise tetracycline detection. After image quality certification, the color information of both the test and control regions was extracted by clicking the “ANALYZE” button in Figure 3E. According to the principle of the paper chip in tetracycline detection, a white color in the test region indicates the appearance of tetracycline, otherwise the color in the test region should be blue, similar to that in the control region. In order to detect tetracycline, the normalized average intensities in the red channel within the test and control regions are I_t_ and I_c_, respectively, and the test to control intensity ratio in red channel is computed as I_t_/I_c_ as shown in Figure 3F.

It is worth noting that the designed application can be used in any Android smartphone, and the optical system can also be combined with different smartphones. (Though since different smartphones have different imaging camera designs, the positions of the lens and the paper chip may need some minor adjusting.) However, because smartphones of different models have various sizes, for a specific smartphone model, a corresponding 3-D printed structure is required in order to integrate the imaging system with the smartphone. We can also design the general 3-D printed structure for different smartphones, however, this not only complicates the mechanical design, but also increases the cost. Since the cost of the image system is rather cheap (the price of the lens is within 100 RMB, and the cost of the 3-D printed structure is around 80 RMB), 3-D printing based small batch fabrication is still affordable, and thus a 3-D printed structure designed for a specific smartphone is preferred due to the simple design and low cost.

In order to test the performance of the handheld inkjet printing paper chip based smart tetracycline detector, both the control (tetracycline free) and the tetracycline solutions with the concentrations of 0.0512 ng/mL, 6.40 ng/mL and 800 ng/mL were added to both the test and control regions, and Figure 4 shows representative captured images and the statistically extracted test to control intensity ratios in red channel from 5 measurements, all of which were ~1, proving that the smartphone based paper chip reader could obtain the test to precisely control intensity ratios in the red channel.

### 3.2. Sensitivity, Specificity and Stability Testing

Prior to practical applications of the handheld inkjet printing paper chip based smart tetracycline detector, its sensitivity, specificity and stability were all tested. PBS solution with different tetracycline concentrations of 0.0102 ng/mL, 0.0512 ng/mL, 0.256 ng/mL, 1.28 ng/mL and 6.40 ng/mL were measured with the proposed smart tetracycline detector, and their corresponding tests to control intensity ratios in the red channel are listed in Figure 5A. In order to determine the detection limit [35], first, test to control intensity ratios in red channel of multiple measurements in a target free condition were computed, and the average signal and the standard deviation were extracted statistically; next, the signal at the detection limit was computed as the average signal plus three times of standard deviation in the target free condition as the dotted line in Figure 5A; finally, according to the experiments, the detection limit of the target could be estimated. According to the results, the detection limit of the proposed smart tetracycline detector was ~0.05 ng/mL, since when the tetracycline concentration was less than 0.0512 ng/mL, the corresponding test to control intensity ratio in red channel was below 1.18 as the threshold (the test to control intensity ratios in red channel of background plus 3 times standard deviation) [36,37,38]. Moreover, the same samples were also measured by ELISA with the results also shown in Table 1, in which “+” represents the positive result and “−” represents the negative result, and Table 1 shows that commercial ELISA kit also reached a similar detection limit of ~0.05 ng/mL.

Moreover, different antibiotics such as ampicillin, penicillin, streptomycin, kanamycin and tetracycline in PBS solution with the same concentration of 64 ng/mL were also detected using the handheld inkjet printing paper chip based smart tetracycline detector, and their corresponding test to control intensity ratios in red channel are listed in Figure 5B. According to the results, the obtained test to control intensity ratios in the red channel of ampicillin, penicillin, streptomycin and kanamycin are all close to ~1 while that of tetracycline is close to 2, indicating that the proposed smart tetracycline detector can distinguish the tetracycline with excellent specificity.

Finally, two extra tests were implemented in order to test the stability of the fabricated paper chips. Paper chips fabricated in 5 batches were used for tetracycline detection, and the results are listed in Table 2, which shows that the paper chips fabricated in different batches could obtain the same detection results. In addition, paper chips fabricated in the same batch but used for tetracycline detection at 0, 2, 4, 6, and 8 days after fabrication (conserved in room temperature) were also used for tetracycline detection, and the results are also listed in Table 2, which shows that the paper chips with different conservation time could still obtain the same detection results. It is believed that both tests proved the stability of the fabricated paper chips.

As the proposed handheld inkjet printing paper chip based smart tetracycline detector could reach the required sensitivity with a rather low detection limit of ~0.05 ng/mL, since it can also distinguish tetracycline with excellent specificity and stability, it could be then adopted in practical applications for tetracycline detections.

### 3.3. Practical Applications

In order to test the performance of the handheld inkjet printing paper chip based smart tetracycline detector in practical applications, tetracycline detection in milk was implemented. First, 3 brands of milk were tested by the handheld inkjet printing paper chip based smart tetracycline detector. According to the test to control intensity ratios in the red channel in Figure 6, no tetracycline was detected, which was also indicated by ELISA in Table 3. In order to prove the measurements in higher precision, all 3 brands of milk were tested by LC-MS, indicating that no tetracycline occurred in the milk samples. Next, extra tetracycline was added into the certificated tetracycline-free milk (Brand 1), and the tetracycline concentrations in the milk were 0.0102 ng/mL, 0.0512 ng/mL, 6.40 ng/mL, 160 ng/mL and 800 ng/mL, respectively. These samples were detected by the handheld inkjet printing paper chip based smart tetracycline detector. With the same detection limit determination method used in Figure 5, its detection limit was still ~0.05 ng/mL according to the results in Figure 6, since when the tetracycline concentration was less than 0.0512 ng/mL, the corresponding test to control intensity ratio in red channel was below 1.26 as the threshold (the test to control intensity ratios in red channel of background plus 3 times standard deviation). Moreover, according to the measurements by ELISA shown in Table 3, the commercial ELISA kit also reached the similar detection limit of ~0.05 ng/mL in milk detecting conditions. According to the practical applications of the handheld inkjet printing paper chip based smart tetracycline detector, it is believed the proposed technique can be well applied in practical applications.

## 4. Conclusions

In this paper, we propose a handheld inkjet printing paper chip based smart tetracycline detector: tetracycline can be determined by inkjet printing paper chip based ELISA testing, and smartphone based paper chip reader and application are designed for automatically determining tetracycline. The smart tetracycline detector has a compact structure with the size of 154 mm × 80 mm × 50 mm and self-supplied internal power, as well as excellent specificity, good stability and high sensitivity with a rather low detection limit of ~0.05 ng/mL that was proved by both PBS solution and milk with tetracycline. The sensitivity of the handheld paper inkjet printing chip based smart tetracycline detector is similar to that of a commercial ELISA kit, but the proposed smart tetracycline detector has a much faster processing speed and lower cost; the whole detection can be finished within 40 min and the cost of the inkjet printing paper chip for single detection is below 1 RMB. Considering its advantages as being compact configuration, high sensitivity, fast speed, simple operation and low cost, it is believed that the proposed handheld inkjet printing paper chip based smart tetracycline detector is a potentially useful tool in antibiotic sensing for routine uses at home and on-site detection in the field.

## Figures and Tables

**Figure 1 micromachines-10-00027-f001:**
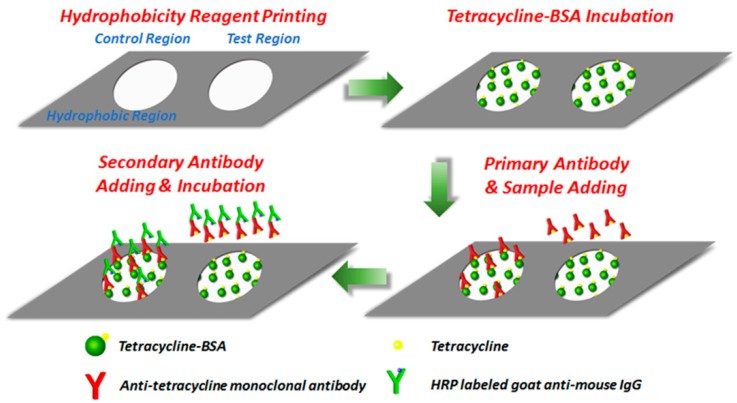
The inkjet printing paper chip based ELISA testing process.

**Figure 2 micromachines-10-00027-f002:**
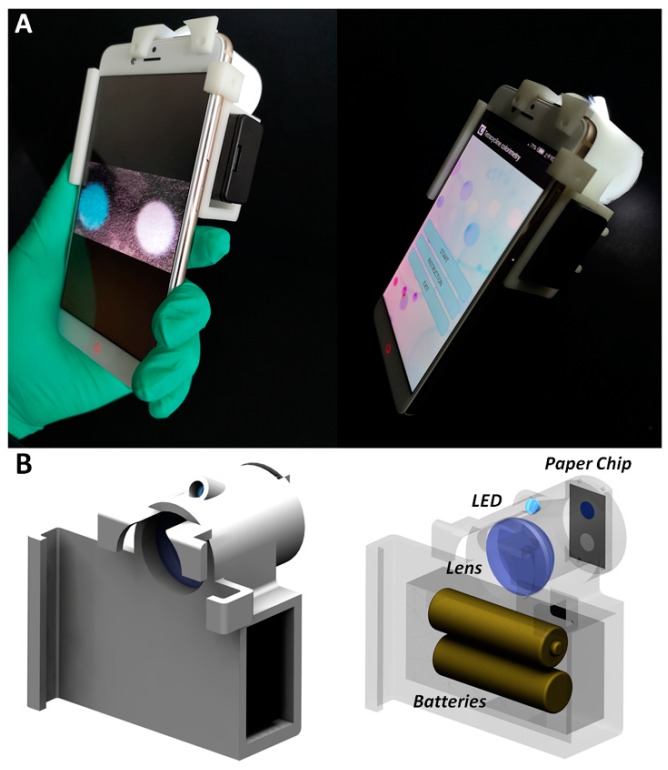
Smartphone based paper chip reader. (**A**) System photo. (**B**) Optical design.

**Figure 3 micromachines-10-00027-f003:**
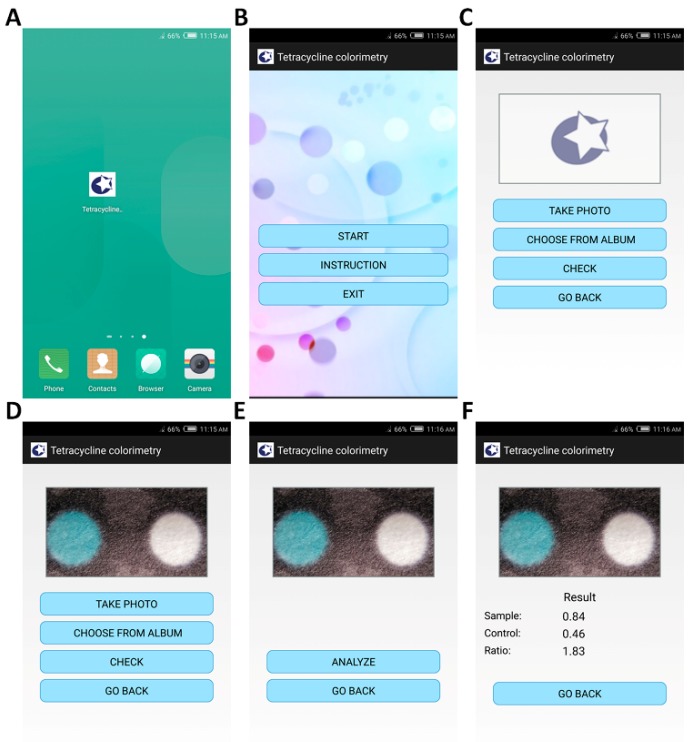
Smartphone application tetracycline colorimetry. (**A**) Application icon on the smartphone desktop. (**B**) Initial interface of the application. (**C**) Inkjet printing paper chip image input. (**D**) Image quality check. (**E**) Image analysis. (**F**) Test to control intensity ratio in red channel extraction.

**Figure 4 micromachines-10-00027-f004:**
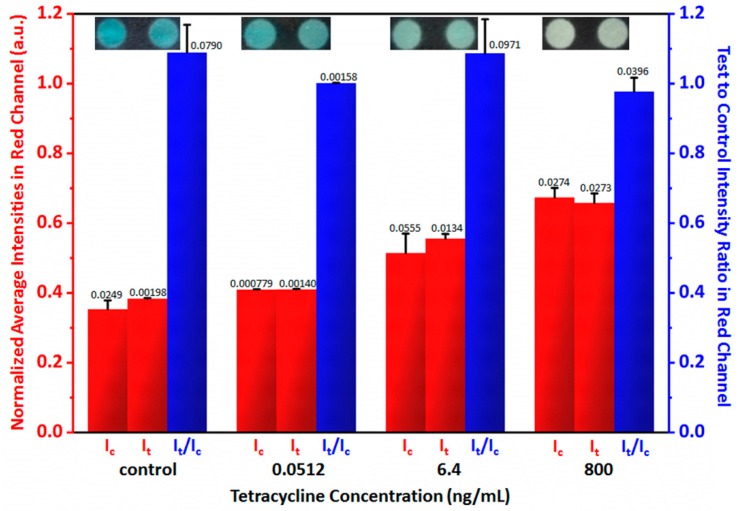
Certification of the handheld inkjet printing paper chip based smart tetracycline detector. The numbers on the columns represent standard deviations computed from 5 measurements.

**Figure 5 micromachines-10-00027-f005:**
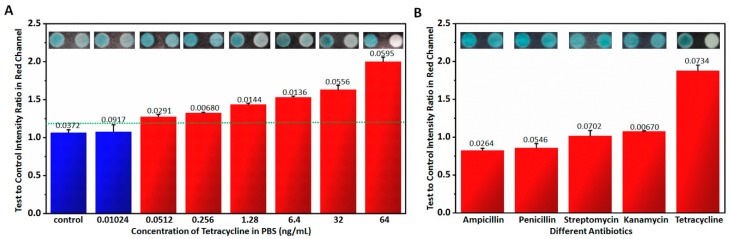
Sensitivity and specificity testing of the handheld inkjet printing paper chip based smart tetracycline detector. (**A**) Sensitivity testing based on detecting different concentrations of tetracycline using the proposed smart tetracycline detector. (**B**) Specificity testing based on detecting different antibiotics using the proposed smart tetracycline detector. The numbers on the columns represent standard deviations computed from 5 measurements. The dotted line in (**A**) indicates the signal at the detection limit.

**Figure 6 micromachines-10-00027-f006:**
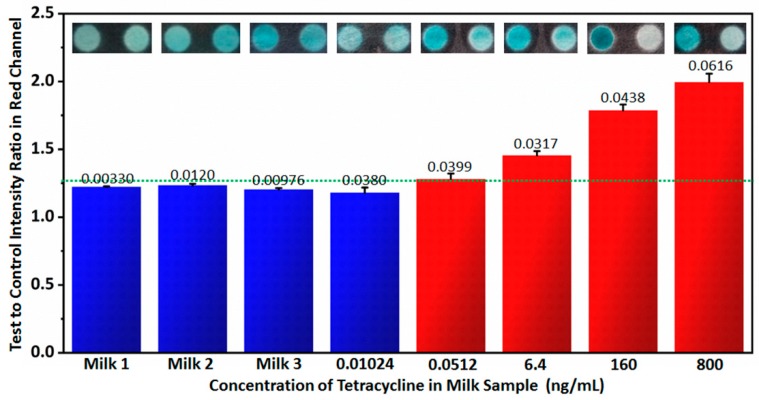
Practical applications of the handheld inkjet printing paper chip based smart tetracycline detector on detecting tetracycline in milk using the proposed technique. The numbers on the columns represent standard deviations computed from 5 measurements. The dotted line indicates the signal at the detection limit.

**Table 1 micromachines-10-00027-t001:** Detection on tetracycline in PBS using the proposed method and commercial ELISA kit.

Sample	Concentration (ng/mL)	Proposed Method (*n* = 5)	ELISA (*n* = 5)
PBS	0	− − − − −	− − − − −
	0.01024	− − − − −	− − − − −
	0.0512	+ + + + +	+ + + + +
	0.256	+ + + + +	+ + + + +
	1.28	+ + + + +	+ + + + +
	6.4	+ + + + +	+ + + + +
	32	+ + + + +	+ + + + +
	64	+ + + + +	+ + + + +

+ positive result; − negative result.

**Table 2 micromachines-10-00027-t002:** Stability testing of the fabricated paper chips.

Tetracycline (ng/mL)	Paper Chips Fabricated in 5 Batches					Paper Chips Conserved with Different Time				
	Batch 1	Batch 2	Batch 3	Batch 4	Batch 5	Day 0	Day 2	Day 4	Day 6	Day 8
0	−	−	−	−	−	−	−	−	−	−
0.0512	+	+	+	+	+	+	+	+	+	+
0.256	+	+	+	+	+	+	+	+	+	+
1.28	+	+	+	+	+	+	+	+	+	+

+ positive result; − negative result.

**Table 3 micromachines-10-00027-t003:** Detection on tetracycline in milk using the proposed method and commercial ELISA kit.

Sample	Concentration (ng/mL)	Proposed Method (*n* = 5)	ELISA (*n* = 5)
Milk	0	− − − − −	− − − − −
	0	− − − − −	− − − − −
	0	− − − − −	− − − − −
	0.01024	− − − − −	− − − − −
	0.0512	+ + + + +	+ + + + +
	6.4	+ + + + +	+ + + + +
	160	+ + + + +	+ + + + +
	800	+ + + + +	+ + + + +

+ positive result; − negative result.

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
