# Peer review of "Handheld Inkjet Printing Paper Chip Based Smart Tetracycline Detector"

_micromachines, 2019, doi:10.3390/mi10010027_

Round 1
Reviewer 1 Report
The manuscript by Jiahao et al. describes a tetracycline detector using a handheld inkjet printing paper chip and a smartphone-based reader. Although the manuscript deals with an important issue for the rapid detection of antibiotics in the field, there is no novelty in constructing the inkjet printing-based paper chip and exploiting smartphone-based colorimetric detection, particularly in the colorimetric detection of tetramethylbenzidine (TMP) based on horseradish peroxidase (HRP) reaction, as well as data collection and analysis using a smartphone-based paper chip reader.
In the Introduction section (line 43), ref. 9 (Analyst 2003, 128, 658) is not adequately cited for the LC-MS method for the detection of tetracycline. Ref. 9 focused on ELISA, not on the LC-MS method.
A chitosan-glutaraldehyde cross-linking method is a special method for antibody immobilization on the paper chip, thus it is not general for the paper chip fabrication. Therefore, an expression in lines 60–64 is overestimated and should be corrected with a new sentence.
In addition, the manuscript addressed a low detection limit of ~0.05 ng/mL tetracycline. However, the obtained value of ~0.05 ng/mL (Fig. 5) is a measured minimum signal, not a detection limit. In general, the detection limit is expressed as the lowest concentration of an analyte that can be detected at a known confidence level, e.g., as S/N ratio times the standard deviation of the blank signal divided by the slope of the calibration curve.
Expressing the results in Figure 6 is confusing. Do the ELISA results indicate conventional ELISA results or your device? If you performed with a conventional ELISA, the detailed method for the commercial ELISA kit should be included in the Materials and Methods section.
Author Response
Reviewer1
The manuscript by Jiahao et al. describes a tetracycline detector using a handheld inkjet printing paper chip and a smartphone-based reader. Although the manuscript deals with an important issue for the rapid detection of antibiotics in the field, there is no novelty in constructing the inkjet printing-based paper chip and exploiting smartphone-based colorimetric detection, particularly in the colorimetric detection of tetramethylbenzidine (TMP) based on horseradish peroxidase (HRP) reaction, as well as data collection and analysis using a smartphone-based paper chip reader.
Thanks for your constructive suggestions.
We have revised the manuscript according to your comments and answered the questions below. We hope these modifications can satisfy your requirements.
The words in red color in the manuscript are revised according to reviewers’ advice and suggestions.
In the Introduction section (line 43), ref. 9 (Analyst 2003, 128, 658) is not adequately cited for the LC-MS method for the detection of tetracycline. Ref. 9 focused on ELISA, not on the LC-MS method.
We have corrected this citing mistake in the revised manuscript, and we have cited a new reference (updated Ref. [9]) on the LC-MSmethod in tetracycline detection.
A chitosan-glutaraldehyde cross-linking method is a special method for antibody immobilization on the paper chip, thus it is not general for the paper chip fabrication. Therefore, an expression in lines 60–64 is overestimated and should be corrected with a new sentence.
According to your suggestion, we have revised the expression to introduce the chitosan-glutaraldehyde cross-linking method in Paragraph 2 of the Introduction Section.
In addition, the manuscript addressed a low detection limit of ~0.05 ng/mL tetracycline. However, the obtained value of ~0.05 ng/mL (Fig. 5) is a measured minimum signal, not a detection limit. In general, the detection limit is expressed as the lowest concentration of an analyte that can be detected at a known confidence level, e.g., as S/N ratio times the standard deviation of the blank signal divided by the slope of the calibration curve.
There are several different "detection limits" that are commonly used, including the instrument detection limit, the method detection limit, the practical quantification limit, and the limit of quantification [1].
In this manuscript, we used the instrument detection limit, and this detection limit is defined as the smallest concentration (or absolute amount) of analyte that has a signal significantly larger than the signal arising from a reagent blank. In order to determine the detection limit, first, test to control intensity ratios in red channel of multiple measurements in target free condition were computed, and the average signal and the standard deviation were extracted statistically; next, the signal at the detection limit was computed as the average signal plus three times of standard deviation; finally, according to the experiments, the detection limit of the target could be estimated.
Besides our work, in many literatures, the detection limit is determined using the same way according to Ref. [2-5].
[1] Long, G.L.; Winefordner, J.D. Limit of Detection A Closer Look at the IUPAC Definition. Anal. Chem. 1983, 55, 712A-724A, doi:10.1021/ac00258a724.
[2] Ren, M.; Xu, H.; Huang, X.; Kuang, M.; Xiong, Y.; Xu, H.; Xu, Y.; Chen, H.; Wang, A. Immunochromatographic assay for ultrasensitive detection of aflatoxin B(1) in maize by highly luminescent quantum dot beads. ACS Appl. Mater. Interfaces2014, 6, 14215-14222, doi:10.1021/am503517s.
[3] Pang, S.; Gao, Y.; Li, Y.; Liu, S.; Su, X. A novel sensing strategy for the detection of Staphylococcus aureus DNA by using a graphene oxide-based fluorescent probe. Analyst2013, 138, 2749-2754, doi:10.1039/c3an36642a.
[4] Hu, J.; Wen, C.Y.; Zhang, Z.L.; Xie, M.; Hu, J.; Wu, M.; Pang, D.W. Optically encoded multifunctional nanospheres for one-pot separation and detection of multiplex DNA sequences. Anal. Chem.2013, 85, 11929-11935, doi:10.1021/ac4027753.
[5] Shi, J.; Chan, C.; Pang, Y.; Ye, W.W.; Tian, F.; Lyu, J.; Zhang, y.; Yang, m. A fluorescence resonance energy transfer (FRET) biosensor based on graphene quantum dots (GQDs) and gold nanoparticles (AuNPs) for the detection of mecA gene sequence of Staphylococcus aureus.Biosens. Bioelectron. 2014,67, 595-600, doi:10.1016/j.bios.2014.09.059.
We have explained the issue in Paragraph 1 of Section 3.2 in the revised manuscript.
Expressing the results in Figure 6 is confusing. Do the ELISA results indicate conventional ELISA results or your device? If you performed with a conventional ELISA, the detailed method for the commercial ELISA kit should be included in the Materials and Methods section.
Figure 6 expresses the results on tetracyclinedetection using the conventional ELISA. In the revised manuscript, we have added detailed process on the conventional ELISA (Section 2.5 Tetracycline detection using Commercial ELISA Kit) in the updated Materials and Methods section.
We hope these modifications can satisfy your requirements.
Thanks again for your advice and suggestions on our research!

Reviewer 2 Report
The Authors reports on a new paper based analysis supported by a hand-handled device connected to a smartphone. The possibility execute ELISA on a paper based chip is well known and also the possibility to exploit smartphone as a point of care is already employed. The only real novelty seem to be the investigation on the detection limit of tetracycline using the described apparatus.
In conclusion this work can be published with a major revision that consist to improve the image e n. 5 A and B with higher resolution plots to give a better evidence of the result, moreover the ELISA comparison with the use of symbol "+" and "-" should be substituted with new plots and better explained in the text.
Author Response
Reviewer 2:
The Authors reports on a new paper based analysis supported by a hand-handled device connected to a smartphone. The possibility execute ELISA on a paper based chip is well known and also the possibility to exploit smartphone as a point of care is already employed. The only real novelty seem to be the investigation on the detection limit of tetracycline using the described apparatus.
Thanks for your constructive suggestions.
We have revised the manuscript according to your comments and answered the questions below. We hope these modifications can satisfy your requirements.
The words in red color in the manuscript are revised according to reviewers’ advice and suggestions.
In conclusion this work can be published with a major revision that consist to improve the image e n. 5 A and B with higher resolution plots to give a better evidence of the result, moreover the ELISA comparison with the use of symbol "+" and "-" should be substituted with new plots and better explained in the text.
In the updated manuscript, we have revised Figures 5 and 6 to provide higher resolution. Moreover, to make the results clear, we have added 2 tables (Tables 1 and 3 in Section 3) to list the detection results based on our method and conventional ELISA. In addition, more explanations on the results have been added in Section 3 of the revised manuscript.
We hope these modifications can satisfy your requirements.
Thanks again for your advice and suggestions on our research!

Reviewer 3 Report
With pleasure, I read your manuscript entitled: “Handheld Inkjet Printing Paper Chip based Smart Tetracycline Detector”. The authors present a smart tetracycline detector based on a handheld inkjet printing paper chip, where tetracycline can be determined by inkjet printing prepared paper chip based ELISA, with the advantages of improved sensitivity, specificity and low cost. The subject is interesting to the readers and researchers. The motivation of the work is clear, and the results are promising.
I have however a few remarks and questions:
- Please check the English writing. There are a few grammar errors along the manuscript. The document must be reviewed by a native or proficient English speaker.
- Section 3.1 – can the app and imaging system be used in any smartphone? It is not clear to me the necessity for the imaging system and its additional cost. Why can’t a common smartphone camera do the same task?
- Please add statistical data to support the experimental tests. How many assays were perform in each test? Add this information to all the plots (4, 5 and 6). Add the standard deviation values to a table (or indicate them in the captions of the figures).
- What was the process to establish the detection threshold? Why test to control intensity ratios in red channel of background plus 3 times standard deviation?
- How are the tests reproducible?
- Please comment the stability of the sensor over time.
- Is there potential to reduce the dimensions of the paper chip (test and control area) and the volumes of the solutions?
Author Response
Reviewer 3:
With pleasure, I read your manuscript entitled: “Handheld Inkjet Printing Paper Chip based Smart Tetracycline Detector”. The authors present a smart tetracycline detector based on a handheld inkjet printing paper chip, where tetracycline can be determined by inkjet printing prepared paper chip based ELISA, with the advantages of improved sensitivity, specificity and low cost. The subject is interesting to the readers and researchers. The motivation of the work is clear, and the results are promising.
I have however a few remarks and questions:
Thanks for your constructive suggestions.
We have revised the manuscript according to your comments and answered the questions below. We hope these modifications can satisfy your requirements.
The words in red color in the manuscript are revised according to reviewers’ advice and suggestions.
Please check the English writing. There are a few grammar errors along the manuscript. The document must be reviewed by a native or proficient English speaker.
We have tried our best to correct the grammar errors in the manuscript, and double checked the manuscript in order to make the manuscript clear and fluent. Moreover, our colleague who has been abroad for many years helped us review the revised manuscript.
Section 3.1 – can the app and imaging system be used in any smartphone? It is not clear to me the necessity for the imaging system and its additional cost. Why can’t a common smartphone camera do the same task?
The designed application can be used in all the Android smartphones, and the optical system can also be combined with different smartphones. (It is worth noting that different smartphones often have different imaging cameras, the positions of the lens and the paper chip may need some minor adjusting.) However, because smartphones of different models have various sizes, for a specific smartphone model, a corresponding 3-D printed structure is required in order to integrate the imaging system with the smartphone. We can also design the general 3-D printed structure for different smartphones, however, it not only complicates the mechanical design, but also increases the cost. Since the cost of the proposed image system is rather cheap (the price of the lens is within 100 RMB, and the cost of the 3-D printed structure is around 80 RMB), 3-D printing based small batch fabrication is still affordable, thus 3-D printed structure designed for specific smartphone is preferred due to the simple design and low cost.
In the revised manuscript, we have explained the application ranges of our designed application and the imaging system, as well as the cost of the system fabrication in Paragraph 3 of Section 3.1.
Please add statistical data to support the experimental tests. How many assays were perform in each test? Add this information to all the plots (4, 5 and 6). Add the standard deviation values to a table (or indicate them in the captions of the figures).
For results listed in Figures 4, 5 and 6, the statistical data was extracted from 5 independent measurements. In the revised manuscript, we have added the repeated detection numbers in the figure caption, and we have also added the standard deviations in the updated Figures 4, 5 and 6.
What was the process to establish the detection threshold? Why test to control intensity ratios in red channel of background plus 3 times standard deviation?
There are several different "detection limits" that are commonly used, including the instrument detection limit, the method detection limit, the practical quantification limit, and the limit of quantification [1].
In this manuscript, we used the instrument detection limit, and this detection limit is defined as the smallest concentration (or absolute amount) of analyte that has a signal significantly larger than the signal arising from a reagent blank. In order to determine the detection limit, first, test to control intensity ratios in red channel of multiple measurements in target free conditions were computed, and the average signal and the standard deviation were extracted statistically; next, the signal at the detection limit was computed as the average signal plus three times of standard deviation; finally, according to the experiments, the detection limit of the target could be estimated.
Besides our work, in many literatures, the detection limit is determined using the same way according to Ref. [2-5].
[1] Long, G.L.; Winefordner, J.D. Limit of Detection A Closer Look at the IUPAC Definition. Anal. Chem. 1983, 55, 712A-724A, doi:10.1021/ac00258a724.
[2] Ren, M.; Xu, H.; Huang, X.; Kuang, M.; Xiong, Y.; Xu, H.; Xu, Y.; Chen, H.; Wang, A. Immunochromatographic assay for ultrasensitive detection of aflatoxin B(1) in maize by highly luminescent quantum dot beads. ACS Appl. Mater. Interfaces2014, 6, 14215-14222, doi:10.1021/am503517s.
[3] Pang, S.; Gao, Y.; Li, Y.; Liu, S.; Su, X. A novel sensing strategy for the detection of Staphylococcus aureus DNA by using a graphene oxide-based fluorescent probe. Analyst2013, 138, 2749-2754, doi:10.1039/c3an36642a.
[4] Hu, J.; Wen, C.Y.; Zhang, Z.L.; Xie, M.; Hu, J.; Wu, M.; Pang, D.W. Optically encoded multifunctional nanospheres for one-pot separation and detection of multiplex DNA sequences. Anal. Chem.2013, 85, 11929-11935, doi:10.1021/ac4027753.
[5] Shi, J.; Chan, C.; Pang, Y.; Ye, W.W.; Tian, F.; Lyu, J.; Zhang, y.; Yang, m. A fluorescence resonance energy transfer (FRET) biosensor based on graphene quantum dots (GQDs) and gold nanoparticles (AuNPs) for the detection of mecA gene sequence of Staphylococcus aureus.Biosens. Bioelectron. 2014,67, 595-600, doi:10.1016/j.bios.2014.09.059.
We have explained the issue in Paragraph 1 of the Section 3.2 in the revised manuscript.
How are the tests reproducible? Please comment the stability of the sensor over time.
The paper chip is use-and-throw, in other words, a paper chip can only be used once for tetracycline detection.
In order to test the stability of the paper chip fabrication, twoextra tests were implemented. Paper chips fabricated in 5 batches were used for tetracycline detection, and the results are listed in Table 2, which shows that the paper chips fabricated in different batches could obtain the same detection results. Besides, paper chips fabricated in the same batch (Dec. 12, 2018), but used for tetracycline detection at 0 (Dec. 12, 2018), 2 (Dec. 14, 2018), 4 (Dec. 16, 2018), 6 (Dec. 18, 2018), and 8 (Dec. 20, 2018)days after fabrication, were also used for tetracycline detection, and the results are also listed in Table 2, which shows that the paper chips with different conservation time could still obtain the same detection results. It is believed that both tests proved the stability of the fabricated paper chips.
In the updated manuscript, we have added Table 2 as well as its explanation in Section 3.2 to express the stability of our fabricated paper chips.
Is there potential to reduce the dimensions of the paper chip (test and control area) and the volumes of the solutions?
If the size of the paper chip is further decreased, though the volume of the solutions for target detection can be reduced, the interval between the control and test regions will be even smaller, cross-interaction inevitably occurs during the sampling adding and washing procedures. While large size of the paper chip not only requires more sample solution, but also leads to inhomogeneous coloration. We optimized the size of the paper chip in this research, and found the used configuration (both the test and control regions with the diameter of 6 mm and the separation of 6 mm) not only had homogeneous coloration, but also avoided the cross-interaction during sample adding and washing, besides, it required relatively low volume of the solution.
In the updated manuscript, we have added more explanation in Section 2.3 to briefly express the size optimization of the paper chip.
We hope these modifications can satisfy your requirements.
Thanks again for your advice and suggestions on our research!

Round 2
Reviewer 1 Report
Thank you for the efforts to revise the manuscript by Jiahao et al.
Compared to the previous version, the revised manuscript improved the quality and presentation of the work, including additional experimental results.
Thus, I would recommend for publication in Micromachines.
Reviewer 3 Report
The authors improved the manuscript and answered my questions and comments. Suggest accept as it is.